# Mitigating Representation Bottlenecks in Multiple Instance Learning

## Abstract

Multiple Instance Learning (MIL) is widely used for Whole Slide Image classification in computational pathology, yet existing approaches suffer from a representation bottleneck where diverse patch-level features are compressed into a single slide-level embedding. We propose *Divide-and-Distill (D&D)*, which clusters the feature space into coherent regions, trains expert models on each cluster, and distills their knowledge into a unified model. Experiments demonstrate that *D&D* consistently improves six state-of-the-art MIL methods in both accuracy and AUC while maintaining single-model inference efficiency.

## 1 Introduction

Whole Slide Images (WSIs) are digital scans of histology slides with gigapixel size and multi-resolution structure [van der Laak et al., 2021]. Their large size makes direct neural network training infeasible, requiring preprocessing such as background removal and tiling into fixed-size patches [van der Laak et al., 2021]. Due to high annotation costs, WSIs are typically analyzed using Multiple Instance Learning (MIL), where each slide is treated as a "bag" of unlabeled patch instances [van der Laak et al., 2021]. In binary classification, a bag is positive if at least one instance is positive.

Attention-based models like ABMIL [Ilse et al., 2018] and CLAM [Lu et al., 2021] improved upon early pooling strategies by weighting patches based on relevance. Recent advances include dual-stream networks [Li et al., 2021], transformer-based approaches [Shao et al., 2021], and structured state-space formulations [Fillioux et al., 2023]. Despite progress, MIL methods face performance plateaus due to representational bottlenecks [Waqas et al., 2024], where aggregation compresses diverse instance-level features into a single slide-level representation.

To address these challenges, we propose *Divide-and-Distill (D&D)*, a framework that partitions the feature space into representation-coherent regions, trains localized expert models on each cluster, and distills this knowledge into a unified global model without increasing inference cost.

## 2 Methodology

Given a WSI for subject $j$, we denote $\mathbf{X}^j = \{\mathbf{x}_1^j, \mathbf{x}_2^j, \ldots\}$ as the set of resulting patches, where each patch $\mathbf{x}_n^j$ represents a small region of the WSI. A feature extractor $f(\cdot)$ compresses each patch into a representative embedding: $\mathbf{z}_n^j = f(\mathbf{x}_n^j)$, yielding the set of embeddings $\mathbf{Z}^j = \{\mathbf{z}_1^j, \mathbf{z}_2^j, \ldots\}$. The patch-level embeddings $\mathbf{Z}^j$ are aggregated using a pooling function $g(\cdot)$ into a WSI-level representation $\mathbf{z}_{\text{WSI}}^j = g(\mathbf{Z}^j)$, which is used to predict the slide label $\hat{Y}^j = \texttt{softmax}(\mathbf{z}_{\text{WSI}}^j)$. However, this aggregation step acts as a *representation bottleneck*: a single vector must summarize thousands of heterogeneous tissue regions, inevitably discarding discriminative local information.

Submitted to 39th Conference on Neural Information Processing Systems (NeurIPS 2025) Workshop: MedEurIPS. Do not distribute.

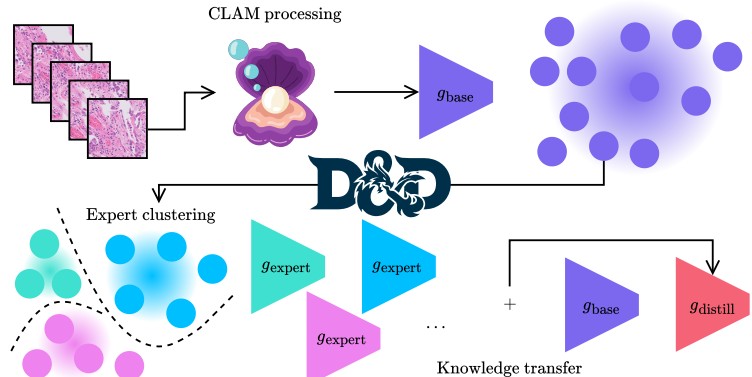

Figure 1: **Overview of the proposed *D&D* framework:** train base model, cluster slide embeddings, train cluster-specific experts, distill into unified model.

## 2.1 Divide-and-Distill (D&D) Framework

To mitigate this bottleneck, we propose *Divide-and-Distill (D&D)*, a method-agnostic framework composed of four stages, which are summarized in Figure 1 and analyzed below.

- **Stage 1: Global base training.** A baseline MIL model $g_{\text{base}}(\cdot)$ is trained on all WSIs using a standard cross-entropy objective function [Hertz et al., 1991], producing slide-level embeddings $\mathbf{z}^j_{\text{WSI}}$ that capture global context.

- **Stage 2: Feature-space partitioning.** The resulting slide representations are clustered into $C$ coherent groups by applying a clustering function $\phi(\cdot)$ (e.g., $k$-means).

- **Stage 3: Expert specialization.** Each cluster defines a subset of WSIs $\mathcal{D}_c = \{(\mathbf{Z}^j, Y^j) \mid \phi(\mathbf{z}^j_{\text{WSI}}) = c\}$, and an expert MIL model $g_{\text{expert},c}(\cdot)$ is trained on each subset $c$ to capture localized tissue patterns and reduce intra-cluster variation.

- **Stage 4: Knowledge distillation.** Knowledge distillation [Hinton et al., 2015] is leveraged to combine the global context captured by the base model with the fine-grained, cluster-specific knowledge of expert models, producing a single model $g_{\text{distill}}(\cdot)$. The objective comprises three components: (1) supervised cross-entropy with ground truth, (2) KL divergence from the base model's predictions, and (3) averaged KL divergence from the $C$ expert models:

$$\mathcal{L} = \mathcal{L}_{\text{CE}} + \lambda_{\text{base}}\mathcal{D}_{\text{KL}}(\hat{Y}_{\text{base}}\|\hat{Y}_{\text{distill}}) + \frac{\lambda_{\text{expert}}}{C}\sum_{c=1}^{C}\mathcal{D}_{\text{KL}}(\hat{Y}_{\text{expert},c}\|\hat{Y}_{\text{distill}}).$$

    This encourages $g_{\text{distill}}$ to retain global discriminative patterns while leveraging cluster-specific expertise from all specialists. We set $\lambda_{\text{base}} = 1$ and $\lambda_{\text{expert}} = 1$ for simplicity.

## 2.2 Information Theory Perspective

We denote $\mathcal{I}(\cdot;\cdot)$ as mutual information. In MIL, aggregation computes a slide-level representation $\mathbf{z}^j_{\text{WSI}} = g(\mathbf{Z}^j)$. By the data-processing inequality, we obtain $\mathcal{I}(\mathbf{Z}^j;Y^j) \leq \mathcal{I}(\mathbf{z}^j_{\text{WSI}};Y^j)$ We define the information-compression loss $\mathcal{L}_{\text{comp}} = \mathcal{I}(\mathbf{Z}^j;Y^j) - \mathcal{I}(\mathbf{z}^j_{\text{WSI}};Y^j)$, which grows in absolute terms when the slide-level representation is overly compressive relative to the heterogeneous patch information. The decomposition in Stage 2 of the *D&D* reduces the effective complexity of each subproblem. For cluster $c$, the local compression loss $\mathcal{L}^{(c)}_{\text{comp}} = \mathcal{I}(\mathbf{Z}^j_c;Y^j_c) - \mathcal{I}((\mathbf{z}^j_{\text{WSI}})_c;Y^j_c)$ satisfies $\mathcal{I}(\mathbf{Z}^j_c;Y^j_c) < \mathcal{I}(\mathbf{Z}^j;Y^j)$ by the subset property, allowing each expert to achieve better local approximations. In Stage 3, each expert focuses on a specific region of the feature space, allowing for specialized pattern recognition without the full complexity of the global problem. The local compression loss for each expert satisfies $\sum_{c=1}^{C}|\mathcal{D}_c|\cdot\mathcal{L}^{(c)}_{\text{comp}} < |\mathcal{D}|\cdot\mathcal{L}_{\text{comp}}$, where $|\mathcal{D}_c|$ and $|\mathcal{D}|$ represent the sizes of cluster $c$ and the full dataset, respectively. In Stage 4, the distilled model approximates the combined information from all sources and hence the mutual information is defined as $\mathcal{I}(\mathbf{Z}^j;Y^j)_{\text{distill}} \approx \max(\mathcal{I}(\mathbf{Z}^j;Y^j)_{\text{base}}, \cup_{c=1}^{C}\mathcal{I}(\mathbf{Z}^j_c;Y^j_c)_{\text{expert},c})$.

| | Method | | CAMELYON-16 | | TCGA-NSCLC | | BRACS | |
|---|---|---|---|---|---|---|---|---|
| | | | ACC | AUC | ACC | AUC | ACC | AUC |
| **ResNet-50** | Mean Pool | | 72.1 | 60.1 | 80.0 | 90.0 | 25.3 | 59.9 |
| | | + D&D | 71.3 ↓0.8 | 60.4 ↑0.3 | 82.5 ↑2.5 | 91.9 ↑1.9 | 36.0 ↑10.7 | 62.2 ↑2.3 |
| | Max Pool | | 81.4 | 80.4 | 81.1 | 90.8 | 35.6 | 71.2 |
| | | + D&D | 79.8 ↓1.6 | 82.9 ↑2.5 | 82.7 ↑1.6 | 91.4 ↑0.6 | 38.0 ↑2.4 | 73.2 ↑2.0 |
| | ABMIL Ilse et al. [2018] | | 78.3 | 77.0 | 81.8 | 90.3 | 35.6 | 70.9 |
| | | + D&D | 82.9 ↑4.6 | 82.1 ↑5.1 | 84.2 ↑2.4 | 91.9 ↑1.6 | 43.7 ↑8.1 | 74.8 ↑3.9 |
| | TransMIL Shao et al. [2021] | | 83.7 | 78.9 | 80.4 | 88.9 | 33.3 | 66.8 |
| | | + D&D | 83.7 ↑0.0 | 80.2 ↑1.3 | 81.2 ↑0.8 | 90.2 ↑1.3 | 35.6 ↑2.3 | 70.3 ↑3.5 |
| | S4MIL Fillioux et al. [2023] | | 80.6 | 84.3 | 82.3 | 90.9 | 37.9 | 73.2 |
| | | + D&D | 78.3 ↓2.3 | 82.9 ↓1.4 | 83.5 ↑1.2 | 91.6 ↑0.7 | 40.2 ↑2.3 | 74.6 ↑1.4 |
| | MambaMIL Yang et al. [2024] | | 76.0 | 78.5 | 81.0 | 89.8 | 41.4 | 73.9 |
| | | + D&D | 77.5 ↑1.5 | 84.2 ↑5.7 | 82.1 ↑1.1 | 91.4 ↑1.6 | 42.5 ↑1.1 | 78.8 ↑4.9 |
| **UNI** | Mean Pool | | 70.5 | 64.7 | 86.5 | 94.4 | 33.3 | 65.9 |
| | | + D&D | 73.6 ↑3.1 | 75.4 ↑10.7 | 87.5 ↑1.0 | 95.2 ↑0.8 | 37.9 ↑4.6 | 67.3 ↑1.4 |
| | Max Pool | | 95.3 | 97.4 | 86.1 | 94.0 | 35.6 | 71.2 |
| | | + D&D | 96.9 ↑1.6 | 98.3 ↑0.9 | 88.4 ↑2.3 | 95.2 ↑1.2 | 42.5 ↑6.9 | 72.6 ↑1.4 |
| | ABMIL Ilse et al. [2018] | | 96.9 | 99.7 | 87.8 | 94.4 | 40.2 | 78.2 |
| | | + D&D | 96.9 ↑0.0 | 99.4 ↓0.3 | 89.2 ↑1.4 | 96.1 ↑1.7 | 46.0 ↑5.8 | 80.9 ↑2.7 |
| | TransMIL Shao et al. [2021] | | 96.9 | 97.8 | 86.3 | 93.0 | 33.3 | 69.7 |
| | | + D&D | 95.3 ↓1.6 | 98.7 ↑0.9 | 87.2 ↑0.9 | 95.1 ↑2.1 | 41.4 ↑8.1 | 76.4 ↑6.7 |
| | S4MIL Fillioux et al. [2023] | | 89.1 | 97.2 | 87.1 | 95.2 | 41.4 | 75.0 |
| | | + D&D | 94.6 ↑5.5 | 99.2 ↑2.0 | 88.4 ↑1.3 | 96.3 ↑1.1 | 48.3 ↑6.9 | 78.9 ↑3.9 |
| | MambaMIL Yang et al. [2024] | | 96.9 | 99.3 | 86.6 | 94.3 | 40.2 | 73.6 |
| | | + D&D | 96.9 ↑0.0 | 99.6 ↑0.3 | 87.7 ↑1.1 | 95.3 ↑1.0 | 42.5 ↑2.3 | 78.2 ↑4.6 |

Table 1: **Performance comparison between baseline MIL methods and their *D&D*-enhanced variants.** D&D improves performance across six MIL methods on three WSI datasets. Upward (↑) and downward (↓) arrows denote performance changes; color intensity reflects the magnitude of variation (green: improvement, gray: minor change, red: decrease).

## 3 Experiments

We evaluate *D&D* on three publicly available WSI datasets: (1) CAMELYON-16 [Ehteshami Bejnordi et al., 2017], (2) TCGA-NSCLC [The Cancer Genome Atlas Research Network, 2019], and (3) BRACS [Brancati et al., 2022]. WSIs are processed using the CLAM [Lu et al., 2021] framework to extract $256 \times 256$ patches at $10\times$ magnification. For feature extraction, we use either ResNet-50 He et al. [2015] pre-trained on ImageNet Deng et al. [2009] or the UNI foundation model Chen et al. [2024]. We consider six representative MIL baselines: Mean Pooling, Max Pooling, ABMIL Ilse et al. [2018], TransMIL Shao et al. [2021], S4MIL Fillioux et al. [2023], and MambaMIL Yang et al. [2024]. Base and expert models are trained with SGD (lr=$1 \times 10^{-4}$, weight decay=$1 \times 10^{-5}$, dropout=0.25) for 200 epochs, embeddings are clustered into $C = 3$ groups using constrained $k$-means Bradley et al. [2000], and the distilled model is trained with Adam for 300 epochs. We report overall accuracy (ACC) and macro-averaged area under the ROC curve (AUC). Table 1 shows the performance of baseline models and their D&D-augmented counterparts across all datasets and feature extractors. Improvements are positive across all datasets and metrics, with the largest gains observed on the BRACS dataset.

## 4 Discussion & Conclusion

To conclude, we propose D&D, a method-agnostic framework that leverages expert clustering and knowledge distillation to enhance the representation learning capacity of existing MIL methods, whilst maintaining inference efficiency. By leveraging expert model clustering and knowledge distillation, *D&D* overcomes limitations of existing MIL approaches. While *D&D* introduces additional training overhead during the expert learning phase, the framework remains lightweight and preserves single-model inference efficiency.

## 87 Potential Negative Societal Impact

While the proposed *Divide-and-Distill (D&D)* framework aims to improve computational pathology models, potential risks arise from biased or overconfident deployment in clinical workflows. As our evaluation relies on publicly available WSI datasets, the models may inherit demographic or acquisition biases that limit generalization across institutions or patient populations. Overreliance on automated predictions without human oversight could lead to diagnostic errors or unfair outcomes. To mitigate these risks, human-in-the-loop review, external validation on diverse cohorts, and adherence to regulatory and ethical standards are essential prior to clinical adoption.

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
