# OpenReview forum: "Mitigating Representation Bottlenecks in Multiple Instance Learning"
_EurIPS.cc/2025/Workshop/MedEurIPS — EurIPS 2025 Workshop MedEurIPS Submission_

### Official Review · Reviewer_NUXv · 2025-10-24
**A simple and effective approach to address representation bottlenecks in MIL**

**Rating:** 5
**Confidence:** 4

**Review:**

The authors propose a “Divide-and-Distill (D&D)” framework to address the bottleneck in representation space for WSI. The approach is intuitive as it clusters slide embeddings into coherent regions, trains local expert models within each, and then distills their combined knowledge into a single global model.

I found the idea both clear and well defined, and the implementation straightforward enough to be adopted in existing MIL pipelines. The authors provide thorough results based on experiments spanning several representative MIL baselines and datasets.

A few areas could be strengthened: it would help to show how sensitive results are to the number of clusters or to visualize what each expert specializes in as it would provide further insight into the method.

Overall, the approach is well defined and well explained, with thorough experimentation results.

---

### Official Review · Reviewer_ehaF · 2025-10-31
**Interesting DL work for pathological imaging**

**Rating:** 8
**Confidence:** 4

**Review:**

This work is closed to medical domain with a particular focus for pathological imaging. They proposed a method call D&D (nice name btw), which try to cluster the features first before distill the knowledge for each of the cluster for classification task. I believe the contribution of this work is solid and it will bring some interesting discussions.

---

### Decision · Program_Chairs · 2025-10-31

**Decision:**

Accept (Poster)

**Comment:**

Both reviewers find the paper relevant and clearly written. The proposed Divide-and-Distill framework is intuitive and well-motivated, with solid experiments. Minor concerns relate to sensitivity analysis and interpretability, but overall, it fits well within the workshop’s scope.